# Epigenetic Modulation of TLR4 Expression by Sulforaphane Increases Anti-Inflammatory Capacity in Porcine Monocyte-Derived Dendritic Cells

**DOI:** 10.3390/biology10060490

**Published:** 2021-05-31

**Authors:** Xueqi Qu, Christiane Neuhoff, Mehmet Ulas Cinar, Maren Pröll, Ernst Tholen, Dawit Tesfaye, Michael Hölker, Karl Schellander, Muhammad Jasim Uddin

**Affiliations:** 1The Brain Cognition and Brain Disease Institute, Shenzhen Institute of Advanced Technology, Chinese Academy of Sciences, Shenzhen-Hong Kong Institute of Brain Science, Shenzhen 518055, China; 2Institute of Animal Science, Animal Breeding and Husbandry, University of Bonn, Endenicher Allee 15, 53115 Bonn, Germany; mpro@itw.uni-bonn.de (M.P.); ernst.tholen@itw.uni-bonn.de (E.T.); Dawit.Tesfaye@colostate.edu (D.T.); michael.hoelker@itw.uni-bonn.de (M.H.); karl.schellander@itw.uni-bonn.de (K.S.); m.uddin2@uq.edu.au (M.J.U.); 3Department of Animal Science, Faculty of Agriculture, Erciyes University, 38039 Kayseri, Turkey; mucinar@erciyes.edu.tr; 4School of Veterinary Medicine, Murdoch University, Murdoch, WA 6150, Australia; 5Department of Medicine, Bangladesh Agricultural University, Mymensingh 2202, Bangladesh; 6School of Veterinary Science, University of Queensland, Gatton, QLD 4343, Australia

**Keywords:** epigenetic, NF-κB, pig, anti-inflammatory response, HDAC inhibitors

## Abstract

**Simple Summary:**

Epigenetic modifications of the genes regulate the inflammation process that includes the DNA methylation and histone acetylation. Sulforaphane is well known for its immunomodulatory properties. Notably, the mechanism of its anti-inflammatory functions involving epigenetic modifications is unclear. This study highlighted the regulatory mechanism of sulforaphane in the innate immunity responses in an acute inflammatory state employ in vivo cell culture model. Porcine monocyte-derived dendritic cells were exposed to LPS with or without sulforaphane pre-treatment for these purposes. Epigenetics modulations of the important genes and regulatory factors were studies as well as the immune responses of the cells were vigorously studied over the period of time. This study deciphers the mechanism of SFN in restricting the excessive inflammatory reactions, thereby, exerting its protective and anti-inflammatory function though epigenetic mechanism.

**Abstract:**

Inflammation is regulated by epigenetic modifications, including DNA methylation and histone acetylation. Sulforaphane (SFN), a histone deacetylase (HDAC) inhibitor, is also a potent immunomodulatory agent, but its anti-inflammatory functions through epigenetic modifications remain unclear. Therefore, this study aimed to investigate the epigenetic effects of SFN in maintaining the immunomodulatory homeostasis of innate immunity during acute inflammation. For this purpose, SFN-induced epigenetic changes and expression levels of immune-related genes in response to lipopolysaccharide (LPS) stimulation of monocyte-derived dendritic cells (moDCs) were analyzed. These results demonstrated that SFN inhibited HDAC activity and caused histone H3 and H4 acetylation. SFN treatment also induced DNA demethylation in the promoter region of the MHC-SLA1 gene, resulting in the upregulation of Toll-like receptor 4 (TLR4), MHC-SLA1, and inflammatory cytokines’ expression at 6 h of LPS stimulation. Moreover, the protein levels of cytokines in the cell culture supernatants were significantly inhibited by SFN pre-treatment followed by LPS stimulation in a time-dependent manner, suggesting that inhibition of HDAC activity and DNA methylation by SFN may restrict the excessive inflammatory cytokine availability in the extracellular environment. We postulate that SFN may exert a protective and anti-inflammatory function by epigenetically influencing signaling pathways in experimental conditions employing porcine moDCs.

## 1. Introduction

The innate immune system is the first line of defense against invading pathogens. In order to detect a microbial attack, the host relies on sentinel cells, such as dendritic cells (DCs) and macrophages. DCs are specialized antigen-presenting cells (APCs) that are involved in regulating immune responses [1]. The recognition and presentation of invasive pathogens by DCs are triggered by microbe-specific motifs known as pathogen-associated molecular patterns (PAMPs), such as lipopolysaccharide (LPS) in the case of Gram-negative bacteria [2,3]. PAMPs are sensed by the coordinated actions of molecules called pattern-recognition receptors (PRRs), such as Toll-like receptors (TLRs) [4]. TLR4 is proven to be an important sensor for LPS [3]. Moreover, LPS-activated TLR4 triggers the mitogen-activated protein kinase (MAPK), nuclear factor-κB (NF-κB), and interferon-related factor (IRF) signaling transduction pathways [4]. As a result, the transcription of immune genes is induced, including cytokines, which are critical for the activation of innate and adaptive immunity and controlling the inflammatory process [4].

Understanding of the epigenetic mechanisms behind the development and differentiation of the immune system has been advanced considerably in recent years [5]. The mechanisms underlying immunomodulation partly depend on the epigenetic regulation of genes related to immune response processes. Epigenetic mechanisms comprise DNA methylation and histone acetylation, which alter gene expression either by hindering the accessibility of chromatin at the CpG dinucleotide or by modifying the nucleosome of DNA [6]. DNA methylation can control gene transcription including miRNAs, which are considered to be the post-transcriptional regulators of genes [6]. Besides this, histone acetylation modifies chromatin structure and can control DNA accessibility to transcription factors and gene expression [6]. Steady-state levels of core histone acetylation result from the balance between the opposing activities of histone acetyltransferases and histone deacetylases (HDACs) [7]. HDACs regulate the suppression of gene transcription via recruitment of methylated CpG [8]. Therefore, inhibition of HDACs results in a general hyperacetylation of histones, which is followed by transcriptional activation of certain genes through the relaxation of chromatin structure. In recent years, sulforaphane (SFN), an isothiocyanate compound derived from broccoli, has become an important natural substance that can potentially inhibit HDAC activity [9]. Furthermore, SFN has been reported to exhibit antioxidative, antimicrobial, anti-inflammatory, and anti-tumoral properties [10,11,12] that make it a key agent in immunology. Adverse mechanisms have been reported to confer beneficial effects on inflammation restriction with pre-treatment of SFN. SFN enhances bacterial clearance by increasing the phagocytic activity of alveolar macrophages and was found to be beneficial against Gram-negative bacteria infection [12,13]. A study has suggested that SFN can inhibit T cell-mediated autoimmune disease in human APCs by impairing expression of the Th17-related cytokines interleukin (IL)-17A, IL-17F, and IL-22 [14]. Notably, these mechanisms include epigenetic changes resulting from the inhibition of HDAC activity.

Although the anti-inflammatory properties of SFN have been reported previously, the epigenetic mechanisms of such an effect are poorly understood. The pig is a very close approximation species to humans in terms of anatomy, genetics, and physiology of immune system to replicate appropriately the condition under investigation and is thought to respond in the same way as humans to microbial infectious disease. In this study, we have used a well-developed cell culture model [15] of porcine monocyte-derived DCs (moDCs) in vitro, which were stimulated with a Gram-negative bacterial component LPS to mimic a state of inflammation. This study evidences that SFN regulates inflammatory cytokine induction through DNA methylation of TLR4 and histone acetylation, thereby protecting porcine moDCs from apoptosis and inflammatory effect caused by LPS. The goal of this study is to provide laboratory evidence that SFN has a potential role to epigenetically modify either the TLR4 or MHC-SLA1-mediated transcription and protein synthesis process of cytokines during acute inflammatory conditions in a cell culture model.

## 2. Materials and Methods

### 2.1. Ethics Statement

The research proposal and ethics were approved by the Veterinary and Food Inspection Office, Siegburg, Germany (ref. 39600305-547/15). A total of six 35-day-old healthy piglets (Pietrain) with no clinical symptoms or serological evidence of influenza and other respiratory or systemic diseases were used for the study. The animals were housed in the accredited barrier-type animal facilities at the teaching and research station of Frankenforst farm, University of Bonn, Germany. The feeding, housing, and husbandry practices of the animals were carried out in accordance with the recommendations of the European Convention for Protection in accordance with the German performance testing guidelines, observing the animal protection law [16].

### 2.2. Generation of moDCs

DCs were derived from cultured porcine monocytes isolated from peripheral blood monocular cells (PBMCs) following the protocol described previously [15,17]. Briefly, PBMCs were isolated from two porcine peripheral blood samples using Ficoll–Histopaque medium (cat. 10771; Sigma, Germany). PBMCs were cultured in Dulbecco’s Modified Eagle Medium (DMEM) (cat. 41966-029; Invitrogen, Darmstadt, Germany) supplemented with 2% fetal bovine serum (FBS) (cat. 10270; Invitrogen, Germany), 500 IU/mL penicillin–streptomycin (cat. 15140; Invitrogen, Germany), and 0.5% fungizone (cat. 15290-026; Invitrogen, Germany) for 4 h. The non-adherent cells were removed by vacuum aspiration and the adherent monocytes were washed twice using pre-warmed (37 °C) DPBS (cat. 14190-094; Invitrogen, Germany). The cleaned monocytes were cultured in RPMI-1640 medium (cat. 21875; Invitrogen, Germany) supplemented with 10% FBS, 1000 UI/mL penicillin–streptomycin, 1% fungizone, 20 ng/mL recombinant porcine (rp) granulocyte-macrophage colony-stimulating-factor (GM-CSF) (cat. 711-PG-010; R&D System, Abingdon, UK), and 20 ng/mL rp interleukin-4 (IL-4) (cat. 654-P4-025; R&D System, UK) for 7 days at 37 °C with 5% CO2. Half of the medium was replaced every 3 days, with the fresh medium supplemented with rp GM-CSF (20 ng/mL) and rp IL-4 (20 ng/mL). After 7 days of incubation, the adherent moDCs were pooled and re-cultured in a new plate after counting for the subsequent assays.


*Cell treatment conditions*


The cell culture and treatment conditions used in this study were described in our previous report [15]. Briefly, moDCs were seeded in a 6-well cell culture plate with 2 × 10^6^ cells/well and cultured in a CO2 incubator at 37 °C for 24–48 h. Cells were first exposed to 10 µM SFN for 24 h. Afterwards, the medium was replaced and 1 µg/mL LPS (cat. # tlrl-3pelps; InvivoGen) was added. The SNF-untreated cells were used as a control or activated with 1 µg/mL LPS. All cells (SNF-pre-treated and LPS-induced (SNF+LPS), control (Con), and LPS-induced only (LPS)) were harvested at 1, 3, 6, 12, and 24 h post-stimulation (ps). The cells were subjected to genomic DNA, total RNA, and protein extraction. Likewise, the cell culture supernatants were also collected at different time points for protein investigation using enzyme-linked immunosorbent assay (ELISA).

### 2.3. mRNA Quantification Using Quantitative Real-Time PCR

Total mRNA from cells was extracted using an miRNeasy Mini Kit (cat. 217004, QIAGEN, Hilden, Germany), and cDNA was synthesized using an miScript II RT kit (cat. 218161, QIAGEN) from cleaned up RNA according to the manufacturer’s protocol. Quantitative real-time PCR (qRT-PCR) was performed in a StepOnePlus Real-Time PCR System (Applied Biosystems, Foster City, CA, USA). Gene-specific primers (Table 1) were designed using the online Primer3 Program (version 0.4.0) [18]. At the end of the PCR, a melting curve analysis was performed to detect the specificity of the PCR. Details of the PCR conditions have been described previously [15,17]. Each experiment was performed in triplicate and each sample was quantified in duplicate (technical replication). Relative mRNA expression was normalized to the average of two housekeeping genes, hypoxanthine phosphoribosyltransferase 1 (HPRT1) and glyceraldehyd-3-phosphat-dehydrogenase (GAPDH). Gene expression was statistically analyzed using the comparative 2^−ΔΔCT^ method [19].

### 2.4. Cytokine and Chemokine Protein Production

For cytokine and chemokine investigation, moDC cell culture supernatants were collected at 0, 1, 3, 6, 12, and 24 h after LPS treatment. Commercially available ELISA kits were used for the quantification of the cytokines’ tumor necrosis factor alpha (TNF-α) (cat. PRA00; R&D Systems, Abingdon, UK) and interleukin-1ß (IL-1ß) (cat. PLB00B; R&D Systems) and the chemokine IL-8 (cat. P8000; R&D Systems) following the manufacturer’s instructions. The optical density (OD) values were measured using a microplate reader (ThermoMax, Ebersberg, Germany) at a wavelength of 450 nm, and the results were calculated according to the manufacturer’s formula.

### 2.5. Western Blotting

Cells were harvested and lysed using the commercial AllPrep^®^ DNA/RNA/Protein Mini kit (cat. 80004; QIAGEN). Equal amounts of cell lysates were loaded and electrophoresed through precast gels, transferred onto nitrocellulose membranes, and confirmed with ponceau S staining [17]. Membranes were incubated with primary antibodies specific for anti-acetylated histones H3 (H3-Ac) (cat. 06-599; Millipore, MA, USA) and H4 (H4-Ac) (cat. 06-866; Millipore, MA, USA) and ß-actin (cat. Sc-47778; Santa Cruz Biotechnology, Heidelberg, Germany). Then, proteins were identified with horseradish peroxidase (HRP)-conjugated secondary antibodies (donkey anti-goat, cat. sc-2020, Santa Cruz Biotechnology for H3 and H4; and goat anti-rabbit, sc-2004, Santa Cruz Biotechnology for ß-actin). Mouse polyclonal anti-ß-actin antibody was used to correct minor differences in protein loading. Finally, the specific signals were detected by chemiluminescence using the SuperSignal West Pico Chemiluminescent Substrate (cat. 34077, Thermo Scientific, Dreieich, Germany). Images were acquired using Quantity One 1-D analysis software (Bio-Rad, Feldkirchen, Germany).

### 2.6. Apoptosis Assay

The moDCs with or without pre-treatment with SFN for 24 h were further treated with LPS for 24 h. Caspase-3 and -9 activities were determined from the cell lysates using the Caspase-3/CPP32 Colorimetric Assay Kit (cat. #K106-25; BioVision, Milpitas, CA, USA) and the Caspase-9 Colorimetric Assay Kit (cat. #K119-25; BioVision, CA, USA) according to the manufacturer’s protocol. An amount of 100 µg of proteins was used for each assay. The samples were measured at 405 nm in a microtiter plate reader (ThermoMax, Ebersberg, Germany).

### 2.7. Methylation Analysis

Genomic DNA was isolated from each treatment group of moDCs using an AllPrep DNA/RNA/Protein Mini Kit (cat. 80004, QIAGEN) according to the manufacturer’s instructions. To analyze the methylation status of CpG motifs, 300 ng of genomic DNA was bisulfite-treated using an EZ DNA Methylation-Direct Kit (cat. D5020, Zymo Research, Irvine, CA, USA) following the manufacturer’s protocol. The promoter region of TLR4 and MHC-SLA1 genes was applied to the online program MethPrimer to appraise the CpG islands [20]. Primer pairs incorporated with the predicted CpG islands were designed using PerlPrimer and Methyl Primer express Software v. 1.0 (Applied Biosystems Inc.) (Table 1) [21]. PCR primer pairs amplified the promoter region of the candidate genes, and the PCR amplification products were purified using the QIAquick PCR purification kit (cat. 28104, QIAGEN). Afterwards, the purified PCR products were subcloned into the pGEM-T easy vector (cat. A1360, Promega, Madison, WI, USA). A total of 4–8 positive clones from each sample were sequenced using the CEQ8000 sequencer system (Beckman Coulter, Brea, CA, USA) with the M13 primers.

### 2.8. Statistical Analysis

In general, the technical replications were averaged. The statistical differences among diversity treatments and time points of gene expressions, cytokine productions, acetylated protein levels, and HDAC activities were evaluated using the SAS software package v. 9.2 (SAS Institute, Cary, NC, USA). For this purpose, the general linear model (GLM) and analysis of variance (ANOVA) statistics were implemented. Moreover, pairwise comparisons of gene expression levels and cytokine productions were performed between the time points and treatment groups using Tukey’s multiple comparisons test. The data were expressed as means ± standard deviations (SD) or least squared means (LS means) + standard error (SE). *p* < 0.05 (* and small letters), *p* < 0.01 (** and capital letters), and *p* < 0.001 (***) were considered as statistically significant.

## 3. Results

### 3.1. SFN Induced Histone Acetylation and Inhibited HDAC Activity

The effect of SFN pre-treatment on HDAC activity was investigated in either unstimulated or LPS-stimulated moDCs, which represent normal and inflammatory states, respectively. The results showed that pre-incubation of moDCs with SFN remarkably suppressed LPS-induced inhibition of HDAC activity (Figure 1A). The protein analysis also indicated that all single SFN and LPS treatment groups significantly deregulated both acetylated H3 and acetylated H4 production (Figure 1B,C). Moreover, SFN pre-treatment enhanced LPS-induced histone H3, but not H4, acetylation (Figure 1B,C).

### 3.2. Promoter Region Methylation of TLR4 Was Inhibited by SFN in LPS-Treated moDCs

In order to understand how SFN epigenetically affects LPS-activated TLR4 signaling, we assessed epigenetic-related DNA methylation in the gene body of the TLR4 gene (Figure 2). The analysis of the DNA methylation pattern was performed with LPS-induced inflammatory moDCs with or without pre-treatment with SFN. We first investigated gene expression under the combined treatment of SFN and LPS. TLR4 gene expression was quantified at 24 h (Figure 2A). Additionally, to elucidate whether TLR4 activation induced by SFN and LPS affects MyD88 signaling, we quantified the effect of combined treatment with SFN and LPS in moDCs by the expression of low and high levels of MyD88 in a time-dependent manner (0, 1, 3, 6, 12, and 24 h post-stimulation (ps)) (Figure 2B). TLR4 mRNA was significantly upregulated in all cases, and the pre-treatment with SFN remarkably enhanced LPS-induced TLR4 expression (Figure 2A). SFN pre-treatment significantly inhibited LPS-induced MyD88 gene expression between 1 and 3 h of LPS stimulation. Surprisingly, SFN significantly enhanced LPS-induced MyD88 gene expression after 6 h of LPS challenge, which remained constant until 24 h (Figure 2B).

To further address whether the alterations in the expression of these two genes were interfered with by epigenetic modification, the DNA methylation status was determined in the presence of SFN and LPS treatment. According to a previous study [15] on alterations in LPS-induced DNMT1 and DNMT3α by SFN, 10 CpG motifs were plotted in the CpG island (Figure 2C) next to the first exon of the porcine TLR4 region. Additionally, the DNA methylation status was examined using bisulfite sequencing in the case of SFN, LPS, and SFN+LPS treatment groups. The results showed that the number of methylated motifs was distinctly higher in the SFN- and SFN+LPS-treated moDCs compared to those in the LPS- and Con-treated moDCs (Figure 2D). LPS partly demethylated the SFN-induced DNA methylation in the SFN+LPS group (Figure 2D). Interestingly, LPS also induced DNA methylation of the TLR4 gene in porcine moDCs (Figure 2D). However, DNA methylation seems to have no direct effect on the expression of TLR4 mRNA in porcine moDCs.

### 3.3. SFN Pre-Treatment Followed by LPS Treatment Restored DNA Methylation in the Promoter Region of MHC-SLA1 Gene

Besides the TLR4 gene, the promoter region and CpG-rich regions of another essential immune mediator gene, MHC-SLA1, were also analyzed for gene expression and DNA methylation (Figure 3). The qRT-PCR results showed that MHC-SLA1 gene expression was significantly upregulated in the SFN-pre-treated moDCs at 0 h and between 6 and 12 h ps (Figure 3A). Interestingly, the pre-treatment with SFN significantly inhibited LPS-induced MHC-SLA1 gene expression between 1 and 3 h of LPS stimulation, whereas LPS-induced expression of the MHC-SLA1 gene was remarkably regained in the SFN-pre-treated group between 6 and 12 h of LPS stimulation (Figure 3A). The DNA methylation patterns of 21 CpG motifs in the MHC-SLA1 gene promoter region were analyzed using bisulfite sequencing after 24 h of SFN and LPS treatment (Figure 3B). All of the treatment groups were found to be more highly methylated compared to the control group (Figure 3C). Notably, the outcomes of the limited samples showed that combined treatment with SFN and LPS suppressed DNA methylation in response to LPS exposure in the MHC-SLA1 promoter region in porcine moDCs (Figure 3C).

### 3.4. SFN Pre-Treatment Inhibited LPS-Induced Cell Apoptosis

To investigate whether apoptosis is involved in LPS-induced porcine moDCs, SFN-induced cell death and the effects of SFN pre-incubation on LPS-induced inflammatory cells were assessed. The results showed that SFN significantly induced caspase-3 and caspase-9 activities (Figure 4A,B). LPS significantly induced caspase-9 gene expression (Figure 4B). Importantly, SFN pre-treatment significantly inhibited caspase-3 and caspase-9 mRNA expression in LPS-induced porcine moDCs (Figure 4A,B). On the other hand, the LPS treatment showed no difference in caspase-3 activity compared to the control group (Figure 4A).

### 3.5. SFN Dynamically Regulated LPS-Induced Nrf2 and STAT3 Gene Expression

The expression of transcription factors (Nrf2 and STAT3) was quantified using qRT-PCR in LPS-stimulated moDCs at 0, 1, 3, 6, 12, and 24 h ps with or without SFN pre-treatment. SFN significantly inhibited LPS-induced upregulation of Nrf2 gene expression at 3 h ps, whereas the effect of SFN on the gene expression of Nrf2 was reversed at 6 h ps with LPS (Figure 5A). On the other hand, SFN significantly inhibited LPS-induced STAT3 gene expression at 1 h ps with LPS, but the expression of STAT3 was significantly upregulated at 6 h ps with LPS (Figure 5B).

### 3.6. SFN Significantly Inhibited LPS-Induced Pro-Inflammatory Cytokine Secretion

The effects of SFN and LPS combined treatment on the induction of pro-inflammatory cytokine production were determined using qRT-PCR and ELISA. The results indicated that SFN pre-treatment significantly downregulated mRNA expression of the inflammatory cytokines tumor necrosis factor (TNF)-α, IL-1ß, IL-8, and IL-6 at 3 h ps with LPS (Figure 6A,C,E,G). However, the expression levels of these genes were upregulated between 6 and 24 h ps with LPS, except in the case of IL-6, which was upregulated at 12 and 24 h ps. Additionally, the protein levels of TNF-α (Figure 6B), IL-1ß (Figure 6D), and IL-8 (Figure 6E) showed that the HDAC inhibitor SFN significantly inhibited the production of these inflammatory cytokines in a time-dependent manner in the case of LPS stimulation.

### 3.7. SFN Dynamically Regulated LPS-Induced CXCL2 and CCL4 mRNA Expression Levels

In order to further clarify the effect of SFN treatment on the LPS-induced inflammatory phenotypic plasticity of moDCs, chemokine genes’ (CXCL2 and CCL4) expression levels were determined at 0, 1, 3, 6, 12, and 24 h ps with LPS. The results revealed that SFN pre-treatment significantly inhibited the expression levels of CXCL2 and CCL4 genes between 1 and 3 h ps with LPS (Figure 7A,B), whereas it restored the expression levels of LPS-induced CXCL2 and CCL4 mRNA between 6 and 24 h ps (Figure 7A,B).

### 3.8. Figures

**Figure 1 biology-10-00490-f001:**
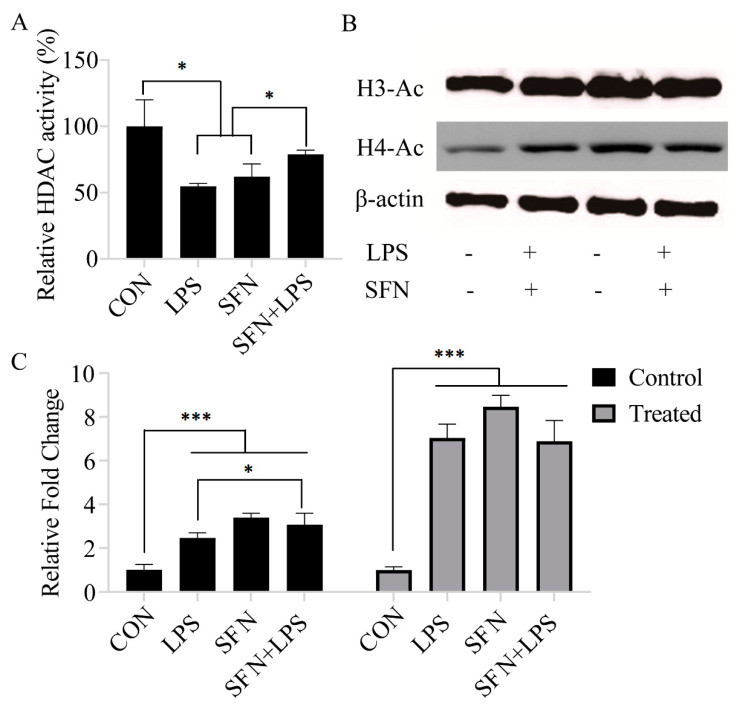
Sulforaphane (SFN) induced histone acetylation and inhibited histone deacetylase (HDAC) activity. The Color-de-Lys HDAC colorimetric activity assay kit was used for global HDAC activity determination. Monocyte-derived dendritic cells (moDCs), cultured for 7 days, were stimulated with lipopolysaccharide (LPS) (1 µg/mL) for 24 h with or without SFN pre-incubation (10 µM) for 24 h (**A**). The results are represented as the mean ± standard deviation (SD) of three independent experiments, and each experiment was performed in duplicate (* *p* < 0.05). The histone acetylation of acetylated H3 and acetylated H4 was measured by Western blotting (**B**). The Western blotting results are from one of three independent experiments. The original image can be found in Appendix A. Quantifications of the immunoblots of acetylated H3 and acetylated H4 are from three independent experiments (* *p* < 0.05; *** *p* < 0.001) (**C**).

**Figure 2 biology-10-00490-f002:**
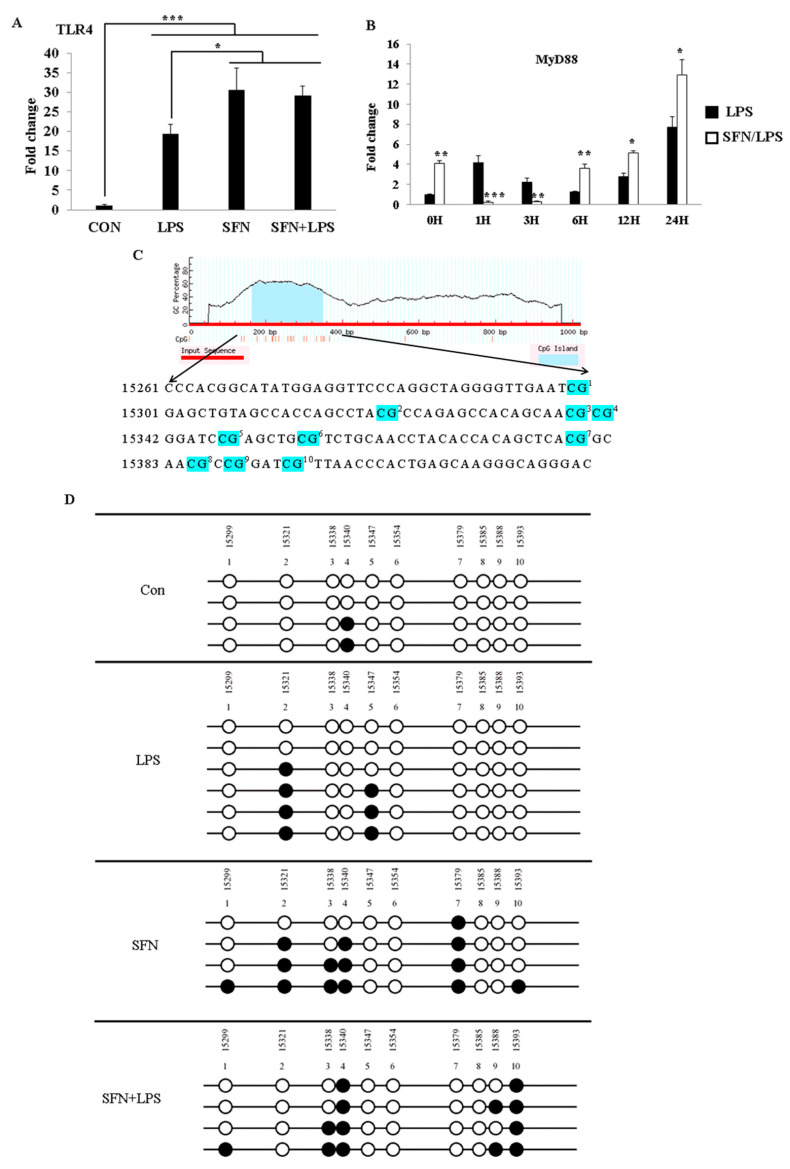
SFN pre-treatment followed by LPS stimulation suppressed DNA methylation of the Toll-like receptor 4 (TLR4) gene. The effects of SFN 10 (µM) on TLR4 and MyD88 gene expression in response to LPS (1 µg/mL) were quantified using qRT-PCR at the indicated time points in moDCs. The moDCs, cultured for 7 days, were pre-incubated with or without SFN and were stimulated with LPS for 0, 1, 3, 6, 12, and 24 h. The TLR4 gene expression was measured at 24 h post-LPS stimulation with or without SFN pre-incubation (**A**). MyD88 mRNA expression was quantified at 0, 1, 3, 6, 12, and 24 h (**B**). The results were combined from three independent experiments, and each experiment was performed in triplicate. The data are represented as the mean ± standard deviations (SD) (* *p* < 0.05; ** *p* < 0.01; *** *p* < 0.001). Ten CpG motifs around the gene body of TLR4 were predicted using the MethPrimer online program (**C**). The DNA methylation status within the CpG island was quantified using bisulfite sequencing PCR (**D**). A minimum of four positive clones were randomly picked for sequencing with M13 primers. The sequencing results were visualized using QUMA software. White plots correspond to unmethylated CpGs, and black plots correspond to methylated CpGs.

**Figure 3 biology-10-00490-f003:**
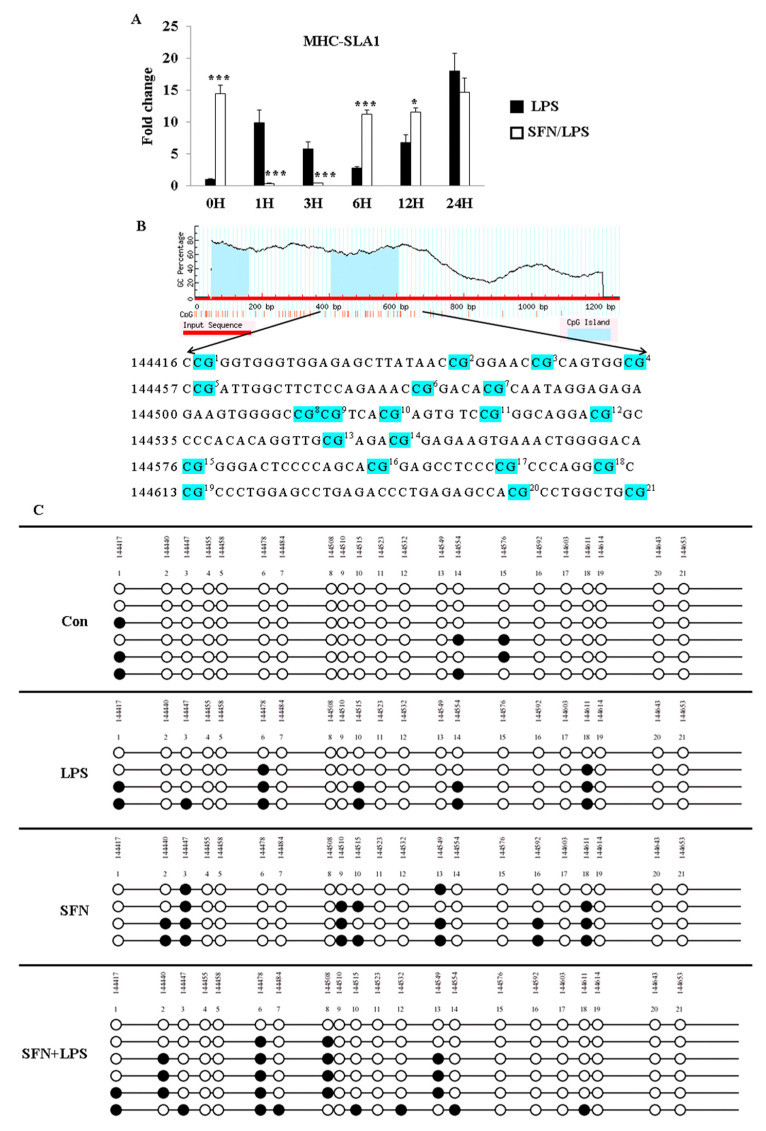
SFN pre-treatment followed by LPS stimulation suppressed DNA methylation of the MHC-SLA1 gene. The effect of SFN on MHC-SLA1 mRNA expression in response to LPS stimulation for 24 h was measured by qRT-PCR (**A**). The results were combined from three independent experiments, and each experiment was performed in triplicate. The data are represented as the mean ± standard deviations (SD) (* *p* < 0.05; *** *p* < 0.001). In total, 21 CpG motifs around the gene body of TLR4 were predicted using the MethPrimer online program (**B**). The DNA methylation status within the CpG island was quantified using bisulfite sequencing PCR (**C**). A minimum of four positive clones were randomly picked for sequencing with M13 primers. The sequencing results were visualized using QUMA software. White plots correspond to unmethylated CpGs, and black plots correspond to methylated CpGs.

**Figure 4 biology-10-00490-f004:**
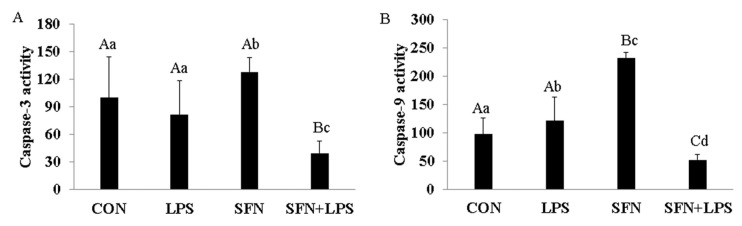
SFN induced caspase-3 and caspase-9 activity and SFN pre-treatment inhibited cell apoptosis. Seven-day-cultured moDCs were used for this experiment. Relative cell apoptotic activity was determined using a Colorimetric Assay Kit. To confirm the induction of caspase-3 and caspase-9 in moDCs, cells were stimulated with LPS for 24 h with or without SFN pre-treatment. Equal amounts of cell lysate were subjected to caspase-3 (**A**) and caspase-9 (**B**) assays. The results were combined from three independent experiments, and each experiment was performed in triplicate. The data are represented as the mean ± standard deviations (SD). Values with different letters denote a significant expression difference among different treated and untreated cells within different groups (small letters: *p* < 0.05; capital letters: *p* < 0.01).

**Figure 5 biology-10-00490-f005:**
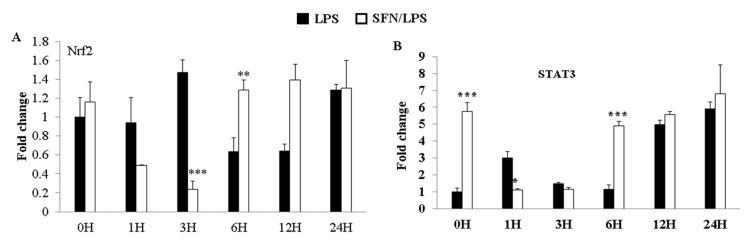
LPS-induced Nrf2 and STAT3 gene expression levels were dynamically regulated by SFN. The effects of SFN on mRNA expression of Nrf2 (**A**) and STAT3 (**B**) in moDCs in response to LPS stimulation for 0, 1, 3, 6, 12, and 24 h were measured by qRT-PCR. The results were combined from three independent experiments, and each experiment was performed in triplicate. The data are represented as the mean ± standard deviations (SD). * *p* < 0.05; ** *p* < 0.01; *** *p* < 0.001.

**Figure 6 biology-10-00490-f006:**
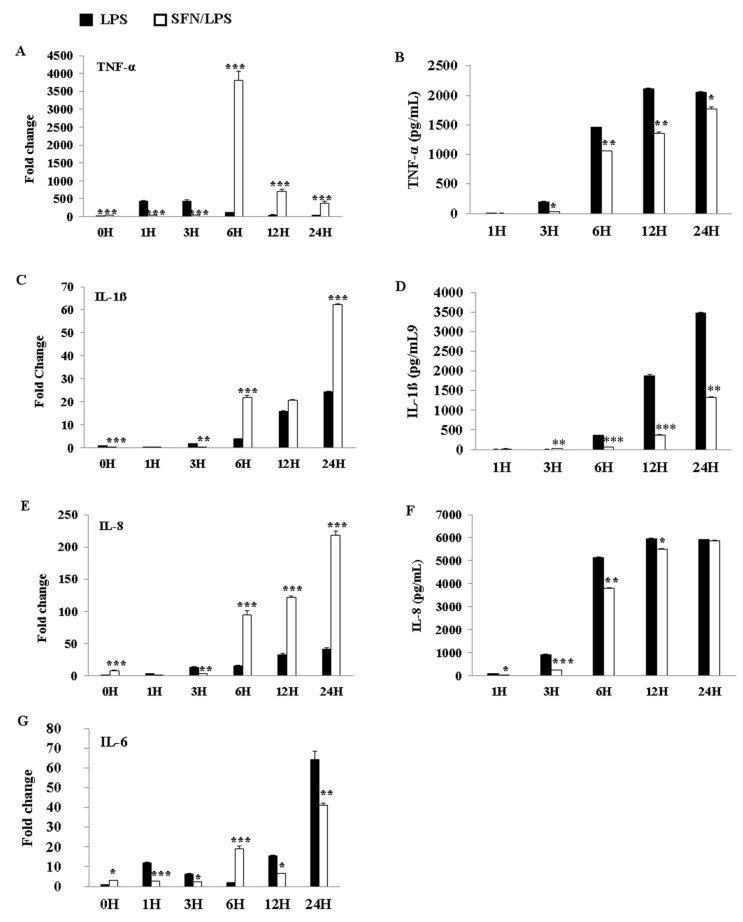
SFN significantly inhibited LPS-induced pro-inflammatory cytokine secretion. The pro-inflammatory cytokine gene expressions of tumor necrosis factor (TNF)-α, interleukin (IL)-1ß, IL-8, and IL-6 were quantified using qRT-PCR. moDCs were stimulated with LPS (1 µg/mL) for 0, 1, 3, 6, 12, and 24 h, with or without SFN pre-incubation (10 M) for 24 h. Total RNA and cell culture supernatants were collected for mRNA and protein measurements, respectively, at 0, 1, 3, 6, 12, and 24 h ps. TNF-α (**A**), IL-1ß (**C**), IL-8 (**E**), and IL-6 (**G**) expression levels in mRNA were quantified by qRT-PCR. The qRT-PCR data of gene expression were combined from three independent experiments, and each experiment was performed in triplicate. TNF-α (**B**), IL-1ß (**D**), and IL-8 (**F**) production levels in protein were measured using ELISA in cell culture supernatants. The ELISA data were combined from two independent experiments, and each experiment was performed in triplicate. The data are represented as the mean ± standard deviations (SD). * *p* < 0.05; ** *p* < 0.01; *** *p* < 0.001.

**Figure 7 biology-10-00490-f007:**
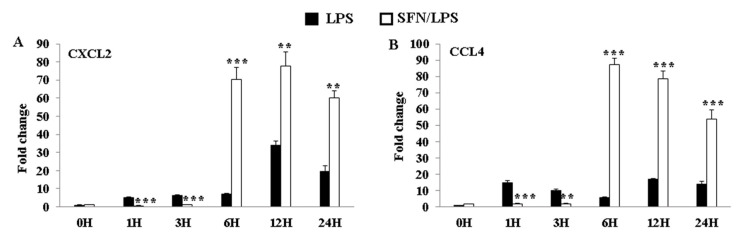
LPS-induced CXCL2 and CCL4 gene expression levels were dynamically regulated by SFN. The effects of SFN on mRNA expressions of CXCL2 (**A**) and CCL4 (**B**) in moDCs in response to LPS stimulation for 0, 1, 3, 6, 12, and 24 h were measured by qRT-PCR. The results were combined from three independent experiments and each experiment was performed in triplicate. The data are represented as the mean ± standard deviations (SD). ** *p* < 0.01; *** *p* < 0.001.

## 4. Discussion

The present study has shown that the HDAC inhibitor SFN has an essential role in the acetylation of histone and non-histone proteins during the regulation of the inflammatory process and innate immune gene expression in porcine moDCs as part of the cells’ defensive response against LPS. SFN induces DNA methylation in both TLR4 and MHC-SLA1 genes. The TLR4 and MHC-SLA1 genes’ transcriptions were epigenetically altered and, thus, suppressed the secretion of inflammatory cytokines in cell culture supernatants, but not in cellular resident cytokines in porcine moDCs. This might be in order to prevent an excessive inflammatory response and might contribute to the resolution of inflammation in porcine moDCs. Notably, in this study, we have shown that SFN-induced DNA methylation of the MHC-SLA1 gene is restored by LPS stimulation. SFN was found to restore the LPS-induced inflammation and innate immune gene expression during the extended period of LPS exposure in porcine moDCs (6 h ps with LPS). Nevertheless, the expression of TLR4/MyD88-dependent genes (such as MD2, MyD88, Nrf2, and STAT3) was strongly inhibited by SFN in the early hours after LPS stimulation.

SFN exhibits potent anti-inflammatory properties that impair the production of inflammatory cytokines in response to LPS by inhibiting LPS engagement with the TLR4/MD2/MyD88 complex and by preferential binding to MD2 in murine bone marrow-derived macrophages [22,23,24]. We know that MyD88 is a core adaptor protein of the TLR4/MD2/MyD88-dependent pathway, which leads to nuclear factor (NF)-κB activation, resulting in pro-inflammatory cytokine expression [25]. Regarding the TLR4 pathway, our present data are consistent with a previous report which showed that SFN suppressed TLR4, MyD88, and NF-κB translocation and thus suppressed the expression of target genes, such as cytokines and chemokines [26]. Interestingly, the LPS-induced TLR4 gene expression was reversed by SFN via delaying of the inflammation process. Consistent with these data, SFN was reported to block initiation of IL-1β and the inflammasome NLRP3 as an anti-inflammatory reagent through inhibition of mitochondrial ROS production [12]. A contradictory conclusion reported that SFN was observed to induce apoptosis and inhibited cell growth in cancer cells by increasing ROS generation and activation of Nrf2 [10]. However, in mammalian cells, ROS have been reported to interact with TLRs, especially TLR2 and TLR4. Therefore, the SFN-induced increase in TLR4 signaling pathway gene expressions at the beginning of LPS treatment may occur through the ROS signaling pathway. Similar results reported that high-dose SFN exposure significantly increased the TLR4 and MyD88 expression levels in endothelial cells, which were suppressed by LPS stimulation [27]. On the other hand, our prior work and that of others have shown that SFN suppressed LPS-induced inflammation via decreased HDAC6 and HDAC10 expression [15]. Although SFN-inhibited HDAC activity has been well studied, given the evidence that SFN increases both global and local histone acetylation in human cells and peripheral blood monocular cells (PBMCs) [28], its anti-inflammatory response mechanism via the HDAC function remains unknown. Chemical inhibition of HDAC6 has been shown to inhibit the formation of the MyD88-TRAF6 signaling complex and represses pro-inflammatory gene expression in macrophages and dendritic cells [15,29]. Negative regulation of the TLR4 signaling pathway by HDAC6 occurred under the mechanism of reversible acetylation of the key adaptor MyD88 [15,28] On the basis of these findings, we postulate that the inhibition of HDAC activity is accompanied by a global increase in histone H3 and H4 acetylation. SFN may induce a difference in the extent of accumulation of acetylated histones in the absence or presence of LPS stimulation. Notably, the distinct classes of HDACs act as either positive or negative regulators of the innate immune response via TLR signaling. Therefore, in this study, the different effects of SFN on HDAC activity indicate that LPS-induced inflammatory moDCs may have a greater ability to resist the downstream effect of HDAC inhibition, thereby accumulating greater amounts of acetylated histones than the control moDCs do [30]. Moreover, the inhibition of LPS-induced HDAC activity may target epigenetic alterations in the early stage of the acute inflammatory process in order to prevent excessive inflammatory cytokine expression. Interestingly, while SFN suppresses TLR4 signaling downstream gene expression levels at the early stage of LPS stimulation, it enhances gene expression at 6 h ps with LPS. However, the ability and molecular mechanism of SFN to epigenetically regulate immune genes in porcine moDCs are largely unknown. We focused on the capability and epigenetic modifications of SFN in LPS-stimulated TLR4 signaling transduction and TLR4-induced cell surface molecule MHC-SLA1 expression. Previously, it has been reported that SFN inhibited LPS-induced TLR4 expression by blocking oligomerization [31]. We found that in the case of SFN pre-treatment, LPS-induced TLR4 gene expression was not always inhibited within 24 h of stimulation [15]. Therefore, we postulate that SFN-induced epigenetic modulations may not only inhibit HDAC enzymes but are also involved in DNA methylation [32]. Additionally, this study aimed to investigate the effects of SFN pre-treatment on the DNA methylation status of TLR4 and MHC-SLA1, either in the promoter or in the gene body, as well as to elucidate how DNA methylation of both genes regulates the relative gene expression in porcine moDCs. For this purpose, the methylation status of the CpG island in the TLR4 gene body in response to LPS stimulation as well as in control moDCs was determined. Notably, porcine TLR4 does not contain the repeat CpG sequence dinucleotide and does not present a typical CpG island (very scarce CpG sites) in the promoter region nor in the first exon. In this study, methylated CpG motifs were detected in the gene body of TLR4, which occurred in both LPS and SFN treatment groups. It was previously reported that LPS-induced TLR4 promoter methylation in human epithelial cells contributes to maintaining homeostasis by regulating mucosal inflammation in the gut [33]. The effects of the inflammation-induced DNA methylation could vary according to different cell types, tissues, species, and stimuli. We found that the alteration of DNA methylation by SFN pre-treatment in the gene body of TLR4 occurred at 24 h ps with LPS. It is necessary to point out that the CpG island of TLR4 is far away from the functional promoter region, and the CpG island does not exist in the first exon. Thus, the SFN- or LPS-induced DNA methylation of TLR4 may not always influence the downstream immune-related gene expression. Therefore, in order to further confirm the epigenetic modulations of SFN on the innate immune response in porcine moDCs, next we investigated another crucial cell surface molecule, MHC-SLA1, and its gene expression. In contrast to the TLR4 gene, the promoter of MHC-SLA1 contains a repeat sequence of CpG motifs and a typical CpG island containing two critical transcriptional sites for NF-κB and TBP (TATA box binding sites). It has been shown that either LPS- or SFN-induced DNA methylation may indirectly inhibit NF-κB and TBP expression in order to prevent excessive inflammatory cytokine production [34]. Porcine MHC-SLA1 is highly polymorphic and has been reported to greatly influence immunological traits [35]. The upregulation of SLA (MHC) in inflammatory immune cells allows for the recognizing of T cells and increasing cytotoxic activity [36]. Therefore, in this study, SFN pre-treatment induced DNA demethylation in the promoter region of MHC-SLA1 and may have a beneficial role in the immune response in the case of LPS stimulation. In agreement with LPS-induced sepsis and inflammatory response, after 24 h of LPS stimulation, SFN-induced CpG sites became demethylated. Together with LPS treatment and inflammatory cytokine overexpression in moDCs, this suggests that active DNA demethylation is involved in inflammation and sepsis induction. A similar result showed that DNA methylation induces SALL4 gene repression in hepatocellular carcinoma in the case of hepatitis virus infection [37]. Our data, together with others, provide evidence that the SFN-induced DNA methylation mechanism plays important roles in acute inflammation.

Furthermore, we have previously shown that the HDAC inhibitor SFN could interfere with the activation of the NF-κB signal transduction pathway [15,38]. LPS-induced TLR4 activation promotes the activation of NF-κB and regulates the expression of inflammatory cytokines, chemokines, and transcription factors [4]. HDAC inhibitors are reported to negatively regulate LPS-induced pro-inflammatory cytokine production in the TLR4 signaling pathway [22,39]. Our study is consistent with previous reports [15]; SFN was found to inhibit TNF-α expression in mRNA in response to LPS. TNF-α repression indicates that SFN may inhibit LPS-induced moDC apoptosis in pig. The restriction of the early production of TNF-α by SFN may contribute to protecting moDCs from excessive TNF-α-induced cell apoptosis. IL-1ß, IL-8, and IL-6 are the principle pro-inflammatory cytokines that mediate the acute inflammatory response. Similar to TNF-α expression, a reduction in the expression of IL-1ß, IL-8, and IL-6 by SFN in LPS stimulated monocyte-derived macrophages in a mouse model [40,41]. Beside pro-inflammatory cytokines, the effect of SFN on chemoattractant (CXCL2 and CCL4) expression was also investigated. CXCL2 and CCL4 are strongly induced by LPS stimulation and recruited by professional antigen-presenting cells to promote cell migration [42]. Similar to the pro-inflammatory cytokines, SFN impaired chemokine expression after stimulation with LPS in porcine moDCs. In agreement with previous reports, these results indicate that the HDAC inhibitor SFN exerts a beneficial role in inflammatory impairment [7,43].

Moreover, SFN is reported to impair inflammatory activity mainly through the activation of the transcription factors Nrf2 and STAT3 (10, 24). Nrf2 is an important modulator and a master transcription factor of antioxidant signaling that serves as a primary cellular defense mechanism to control anti-inflammatory and antioxidant genes [44]. Moreover, SFN promotes Nrf2 activation against inflammation [45,46], and this activation may be due to SFN-induced Nrf2 demethylation [47]. STAT3 is activated in response to various antigens, including LPS, and its constitutive activation directly contributes to inflammation. Previous studies reported that SFN-induced apoptosis via ROS-dependent Nrf2 activation is due to STAT3 phosphorylation [48,49,50]. In agreement with previous studies, we were also able to show a relationship between the inhibition of pro-inflammatory cytokine expression and the activation of Nrf2 and STAT3 by SFN in response to LPS stimulation [11]. These results indicated that the anti-inflammatory activity of SFN is not only modulated by SFN-induced epigenetic alteration, but it also activates Nrf2 and ROS pathways. Previously, we reported that SFN pre-treatment increased the phagocytosis of moDCs when challenged with LPS as a beneficial immune defense mechanism and markedly increased the cell viability [15]. Within LPS, SFN inhibited the immune gene expression including transcription factors, cytokines, and chemokines in antigen-specific (TLR4 pathway) and non-specific (MHC pathway) pathways. However, SFN-induced histone acetylation and DNA demethylation at the gene body or promoter region might suppress important immune genes activation.

This study found that the protein production of the pro-inflammatory cytokines TNF-α, IL-1ß, and IL-8 was markedly suppressed in cell culture supernatants throughout the 24 h LPS stimulation. Consistent with our results, it has been reported that SFN pre-treatment suppresses pro-inflammatory cytokine (TNF-α, IL-1ß, IL-6) secretion levels through the Nrf2 pathway [11]. Inflammation-induced TNF-α has cytotoxic effects on immune cells and stimulates numerous inflammatory mediators, including IL-1ß, IL-8, and IL-6, which are critical for inflammation and tissue damage [51]. This TNF-α induction during acute inflammatory infection may determine whether the cytokine is protective/beneficial in the case of SFN treatment. The results of an in vivo experiment reported that excessive secretion of TNF-α results in severe inflammation and causes early death in a mycobacterial infection mouse model [52]. Clinically, classical HDAC inhibitors (such as TSA and SAHA) have been used as therapeutic agents in inflammatory diseases to suppress inflammatory cytokines [53,54]. Additionally, similar results were previously found in the NF-κB subunits p50 and p65 [15]. Therefore, the present study postulates that SFN may regulate the induction of pro-inflammatory cytokines, either in a culture medium or in cell lysates, at different time points in the LPS-induced inflammatory process, which may be beneficial for porcine moDCs.

## 5. Conclusions

In conclusion, the present data demonstrated that SFN not only induced histone acetylation but also changed the DNA methylation pattern to regulate the expression of immune genes in response to LPS in moDCs. Inhibition of HDAC activity through histone acetylation and the changes in the DNA methylation, together, regulate the immune functions of moDCs in a time-dependent manner during infection. Additionally, the SFN-induced anti-inflammatory response includes activation of the Nrf2-dependent pathway and apoptotic mechanism. Thus, this study postulates that SNF may epigenetically regulate the development of inflammation by modulating innate immune responses, which may point to HDAC inhibitors as potential anti-inflammatory therapeutic agents in infection.

## Figures and Tables

**Table 1 biology-10-00490-t001:** List of primer sequences used in this study.

Gene	Primer Set	Anneal Temperature (°C)	Amplicon Size (bp)	GenBank Accession Number
*TLR4*	F:ATCATCCAGGAAGGTTTCCAC R:TGTCCTCCCACTCCAGGTAG	58	235	NM_001097444.1
*MyD88*	F:CCAGTTTGTGCAGGAGATGA R:TCACATTCCTTGCTTTCGAG	60	185	NM_001099923.1
*MHC-SLA1*	F:AGAAGGAGGGGCAGGACTAT R:TCGTAGGCGTCCTGTCTGTA	60	199	NM_001097431.1
*Nrf2*	F:GTGCCTATAAGTCCCGGTCA R:ATGCAGAGCTTTTGCCCTTA	60	108	XM_003483682.1
*STAT3*	F:ATGCTGGAGGAGAGAATCGT R:AGGGAATTTGACCAGCAATC	60	159	XM_005668829.1
*TNF-α*	F:CCACCAACGTTTTCCTCACT R:CCAAAATAGACCTGCCCAGA	60	247	NM_214022.1
*IL-1ß*	F:GTACATGGTTGCTGCCTGAA R:CTAGTGTGCCATGGTTTCCA	59	137	NM_001005149.1
*IL-6*	F:GGCAGAAAACAACCTGAACC R:GTGGTGGCTTTGTCTGGATT	58	125	NM_214399.1
*IL-8*	F:TAGGACCAGAGCCAGGAAGA R:CAGTGGGGTCCACTCTCAAT	60	174	NM_213997.1
*CXCL2*	F:ATCCAGGACCTGAAGGTGAC R:ATCAGTTGGCACTGCTCTTG	60	152	NM_001001861.2
*CCL4*	F:CTCTCCTCCAGCAAGACCAT R:CAGAGGCTGCTGGTCTCATA	60	191	NM_213779.1
*HPRT1*	F:AACCTTGCTTTCCTTGGTCA R:TCAAGGGCATAGCCTACCAC	60	150	NM_001032376.2
*GAPDH*	F:ACCCAGAAGACTGTGGATGG R:ACGCCTGCTTCACCACCTTC	60	247	AF017079
*TLR4-met-nest*	F:GTATATGGAGGTTTTTAGGTTAGGG R:TCCCTACCCTTACTCAATAAATTAAC	55	153	AY753179
*MHC-SLA1-met-nest*	F:GTTTGGGGAGAAGTTGAGTAGAGT R:AAAAAACAAAAACAAAACAAAATCC	58	293	AJ251829.1

F: Forward primer; R: Reverse primer; bp: base pair.

## Data Availability

The data presented in this study are available within the article.

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
