# Peer review of "Epigenetic Modulation of TLR4 Expression by Sulforaphane Increases Anti-Inflammatory Capacity in Porcine Monocyte-Derived Dendritic Cells"

_biology, 2021, doi:10.3390/biology10060490_

Round 1

Reviewer 1 Report

Overall, I thought this was a very well-written and interesting manuscript.  I would consider shortening the discussion.   

Not sure if there is another way to present Figure 2, but it is cumbersome and hard to decipher the importance of the figure.  

Reviewer 2 Report

In this work, the authors explored the epigenetic effects of sulforaphane on the homeostasis of innate immunity during acute inflammation.

The paper is generally well written and the presented data quite solid. And now comes the famous however.

In several places, the authors overstate their findings eg the last sentence in the introduction. This reviewer does not agree with the claim that the study provides evidence that SFN epigenetically modifies the transcription of cytokines ..... The authors should soften their conclusions by specifying the genes studied. 

The biggest concern is however about the methylation analyses and the presentation of the HDAC blotting.

Fig. 1B looks very artificial. To support this analysis, the authors should provide the full Western blots used for Fig. 1B. The differences in backgrounds raise my skepticism regarding the accuracy of the quantification in Fig. 1C.

The DNA methylation was analysed on a fairly low number of subcloned PCR products by using bisulfite sequencing. The sensitivity of the analysis is low thus questions the reproducibility. The authors may consider to re-analyse the methylation with a more reliable method such as pyrosequencing.

Reviewer 3 Report

The study aimed to determine selected mechanisms of epigenetic modulation of TLR4 that is the member of Toll-like receptors, involved in innate immune system. Investigations were conducted using ethical guidelines for  care and use of animals for research and were approved by Veterinary and Food Inspection, Germany.

TLR4 recognizes LPS and LPS-activated TLR4 triggers the mitogen-activated protein kinase (MAPK), nuclear factor-kB and interferon-related factor signalling transduction pathway. As the result of its activation  transcription of genes  (not immune genes - please improve it and precise). The genes are involved in cytokine producion whch are important for activation both innate and adaptive immunity and i.a. controll the process of inflamation. The Authors focused on epigenetic mechanisms involved in the modulation of TLR4 expression, including the role of sulforaphane which exhibit anti-inflamatory, anti-tumoral and anti-microbial properties. The Introduction section provides interesting info on the consequences of TLRs activation and concerns also epigenetic mechanisms possibly underlying immunomodulation. It is fine. I would suggest however to rebuil the last paragraph of the Introduction section and provide convinced  rationale for the study, including the scientific hypothesis. Hypothesis does'nt exist. The part of the Introduction on lines 82-84  inculdes conclusion (lines 82-84) and should be placed in the Discussion section. 

The Material and Methods - in my opinion the methodology is sufficiently robust. The Materials and Methods  provide sufficient information to replicate the studies and experiments and the experiments are clearly described. Hovewer, at the begining of each sub-chapter I would suggest to add  the goal and  rationale for used methods. It would help the reader to follow. Moreover, please provide rationale for cell treatment conditions. Why SFN at 10 μM for 24 h was used? The results are clearly presented, but statistical analyses should be described more precisely. Please indicate the treatment groups, please indicate point by point what comparisons have been performed. Please do not interpret the results in the Results section, please describe / present them here but discuss them in the Discussion section. The Discussion section is the weakness part of the paper as it is overstate. Moreover, the results do not always justify the interpretations and conclusions. Unfortunately, the quality of the writing in the Discussion section  is not suitable for publication. This is the major weaknesses that need to be corrected prior to consideration for publication.

Reviewer 4 Report

The article: Epigenetic Modulation of TLR4 Expression by Sulforaphane Increases Anti-Inflammatory Capacity in Porcine Monocyte-Derived Dendritic Cell "describes a series of in vitro experiments to prove the anti-inflammatory properties of sulforaphane. Research was conducted on Porcine monocyte-derived dendritic cells. Although many analyzes have been performed: (Westen Blot analysis, qPCR, CpG methylation analysis, caspase activity analysis), the work lacks consistency. At times it is difficult to understand for what purpose the analysis was done. In my opinion, the obtained results are often ambiguous, for example, Fig 1 shows that SFN alone has a similar effect on HDAC activity like LPS, and cells pretreated with SFN did not decrease the activity level of H3 or H4 to the control level. Moreover, why there is no control in Figures 2B, 3A, 5A , B. Why there is so strong overexpression o0f MHC-SLA at 0h point in SFN / LPS group? Perhaps presenting the summary of the results in the table or on a diagram would help to better illustrate the line of reasoning presented in the discussion. In addition, the introduction should explain why the research was carried out on porcine monocyte-derived dendritic cells and to which group of substances Sulforaphane belongs and where it occurs in nature.

Round 2

Reviewer 2 Report

No new data has been added, but the authors have improved the text of the manuscript. I think the paper can be published as it is.

Reviewer 3 Report

I revised the report and in my opinion the manuscript can be accepted in the present form now. The authors revised the manuscrirpt based on my comments. 

Reviewer 4 Report

I could not find the attachments and reply for my comments